# Measurement and Correlation of Solubility of Thiamine Nitrate in Three Binary Solvents and Thermodynamic Properties for the Solutions in Different Binary Mixtures at (278.15–313.15) K

**DOI:** 10.3390/molecules28135012

**Published:** 2023-06-27

**Authors:** Xinda Li, Zhengjiang Wang, Jing Wang, Jiaqi Lu, Jin Mao, Dandan Han, Kangli Li

**Affiliations:** 1State Key Laboratory of Chemical Engineering, School of Chemical Engineering and Technology, Tianjin University, Tianjin 300072, China; lixinda@tju.edu.cn; 2Xi’an TPRI Water-Management & Environmental Protection Co., Ltd., Xi’an 710054, China; wangzhengjiang@tpri.com.cn (Z.W.); wangjing@tpri.com.cn (J.W.); 13299080128@163.com (J.L.); maojin@tpri.com.cn (J.M.); 3Institute of Shaoxing, Tianjin University, Shaoxing 312300, China

**Keywords:** thiamine nitrate, solubility, thermodynamic, polarity

## Abstract

The solubility of thiamine nitrate in {(methanol, acetone, isopropanol) + water} solvents will provide essential support for crystallization design and further theoretical studies. In this study, the solubility was experimentally measured over temperatures ranging from 278.15 to 313.15 K under atmospheric pressure using a dynamic method. The solubility increased with increasing temperature at a constant solvent composition. The dissolving capacity of thiamine nitrate in the three binary solvent mixtures at constant temperature in the low ratio of water ranked as water + methanol > water + acetone > water + isopropanol generally. Interestingly, in the high ratio of water systems, especially when the molar concentration of water was greater than 0.6, the dissolving capacity ranked as water + acetone > water + methanol > water + isopropanol. Additionally, the modified Apelblat equation, *λh* equation, van’t Hoff equation and NRTL model were used to correlate the solubility data in binary mixtures. It turned out that all the selected thermodynamic models could give satisfactory results. Furthermore, the thermodynamic properties of the dissolution process of thiamine nitrate were also calculated based on the modified van’t Hoff equation. The results indicate that the dissolution process of the thiamine nitrate in the selected solvents is all endothermic.

## 1. Introduction

Vitamins are essential for the development and normal growth of human and animal bodies, and lack or excess of them may cause serious physiological problems [1]. In addition, water-soluble vitamins are available in many pharmaceutical dosage forms, such as drinks, tablets, gelatin capsules and syrups [2]. The widespread use of vitamin preparations has stimulated research on accurate, efficient and easy methods for quality control. Thiamine nitrate (C_12_H_17_N_5_O_4_S, CAS Registry No. 532-43-4, Figure 1) is a member of the vitamin B family, which seems to support the axoplasmic transport by supplying energy as ATP. It was found that the distribution of thiamine nitrate in nerve cells is primarily located in membrane structures and has an important effect on the regeneration of damaged cells due to its important role in glucose metabolism of nervous tissue [3,4]. Thiamine nitrate can not only be used as medicine, but also be used as a food fortification agent or feed additive.

During the past years, increasing evidence of the pharmacological effects of thiamine nitrate has been accumulated in experimental and clinical trials [3,5], but only a few studies have attempted to establish purification methods to obtain products with high purity, yields, bulk density and flowability. Due to the polarity of its molecules, the morphology of thiamine nitrate is often like a needle or rod. However, crystal morphology plays a crucial role in chemical and pharmaceutical manufacturing as it impacts solid properties, including bioavailability for pharmaceuticals, dissolving rate, flowability and so on [6,7,8]. It also affects processes such as filtration, drying and compaction [9,10]. Therefore, to produce thiamine nitrate with desirable morphology in the pharmaceutical industry, it is crucial to produce thiamine nitrate in different solvent systems via crystallization as solvents may have a big influence on the morphology of crystals. Moreover, solubility is also needed in order to control the supersaturation, particle size and crystal form in the crystallization process [11]. In our previous work, the solubility of thiamine nitrate in water + ethanol binary solvent mixtures and in aqueous solution with various pH values was measured. However, the morphology of thiamine nitrate in the solutions did not change obviously; therefore, we have to screen other solvent systems [12].

To the best of our knowledge, the solubility of thiamine nitrate in the binary solvent mixtures, including water + methanol, water + acetone and water + isopropanol, has not been reported in the literature, so the solubility data are essential data to produce a thiamine nitrate product with desirable morphology and high yields. In this work, the solubility of thiamine nitrate in these binary solvent mixtures was measured over the temperature range of 278.15 to 313.15 K via a dynamic method at atmospheric pressure (*p* = 0.1 Mpa). In order to extend the applicability of the solubility, the experimental solubility was correlated using the modified Apelblat equation, *λh* equation, van’t Hoff equation and NRTL model, respectively. In addition, the thermodynamic properties of the thiamine nitrate dissolving in different binary solvent mixtures, including the enthalpy, the entropy and the Gibbs energy, were calculated and discussed. Additionally, to ensure that the crystal form remains constant during the experimental process, the identification of the thiamine nitrate crystal form was verified by using powder X-ray diffraction (PXRD).

## 2. Results and Discussion

### 2.1. X-ray Powder Diffraction Analysis

The X-ray powder diffraction (PXRD) pattern verified the identity and the high crystallinity of thiamine nitrate used in this research, and it was found that the PXRD pattern of all the samples remained constant. One sample (*T* = 298.15 K, water + methanol ($x_{1}^{0}\;$ = 0.1, 0.3, 0.5, 0.7, 0.9) was discussed and shown in Figure 2a. It confirmed that the samples did not show any polymorphism or solvates and were not amorphous during the dissolution process. In addition, the simulated XRD pattern sourcing from the single-crystal XRD result is shown in Figure 2b. All the characteristic peaks of the single-crystal XRD pattern can be found in the X-ray powder diffraction pattern (Figure 2a), showing that the structure of the prepared compound was similar to the single-crystal thiamine nitrate.

### 2.2. TGA/DSC

The thermal analysis (TGA/DSC) of thiamine nitrate is presented in Figure 3. It can be seen that the samples decompose before showing a melting characteristic. Therefore, the melting temperature $T_{m}$ of thiamine nitrate cannot be obtained using this conventional calorimetric method. Thus, in this work, the group contribution method [13] was used to estimate the melting point of thiamine nitrate. At equilibrium, the free energy of transition is equal to zero. So, the melting point of organic compounds can be calculated using Equations (1)–(3) as follows:(1)$$\;T_{m}=\Delta H_{m}/\Delta S_{m}$$
(2)$$\Delta H_{m}=\sum n_{i}m_{i}$$
(3)$$\Delta S_{m}=C-Rln\sigma +Rn\emptyset \;$$
where $n_{i}m_{i}\;$ is the contribution of group *i* to the fusion enthalpy, *σ* represents the number of positions into which a molecule can be rotated that are identical with a reference position and ∅ indicates the molecular flexibility which can be calculated using Equation (4).
(4)$$\emptyset =2.435{\;}^{SP3+0.5SP2+0.5RING-1}\;$$

SP3 is the number of sp3 chain atoms; SP2 is the number of sp2 chain atoms and RING represents the number of independent single, fused or conjugated ring systems [14]. The melting point $\;T_{m}$ of thiamine nitrate and the fusion enthalpy $\Delta H_{m}$ calculated from the above multilevel scheme are 486.56 K and 35.13 kJ·mol^−1^, respectively. It is worthwhile to mention that the results were only used for prediction but not as a result of the real physical properties of thiamine nitrate.

### 2.3. Experimental Data

The molar fraction solubility of thiamine nitrate in the binary solvent mixtures, including methanol + water, acetone + water and isopropanol + water, is listed in Table 1, Table 2 and Table 3 and plotted in Figure 4, Figure 5, Figure 6, Figure 7, Figure 8 and Figure 9. It was found that the solubility increased with increasing temperature at a constant solvent composition in all binary systems. In addition, the dissolving capacity rankings of thiamine nitrate in the binary solvent mixtures at a constant temperature were methanol + water > acetone + water > isopropanol + water at the low ratio of water, which may be in conformity with the empirical rule “like dissolves like” [15,16]. As the polarity of the various solvents follows the order of water > methanol > acetone > isopropanol, and the thiamine nitrate is a polar molecule, it can be preferentially dissolved in polar solvents such as water, which is consistent with the solubility. However, with the increase in water, as shown in Figure 4, Figure 5 and Figure 6, it can be easily seen that in methanol + water solvent mixtures, the solubility of thiamine nitrate reached its maximum point at $x_{1}^{0}$ = 0.3 and then decreased obviously with the decrease in water, and the maximum point did not change with temperature; this phenomenon is also called cosolvency, which was described in previous studies [17,18]. The occurrence of these maxima has a complex thermodynamic basis, influenced by both enthalpy and entropy effects, and no definite explanation has been achieved [19]. On the whole, it may be due to the physical–chemical properties of the solvent, such as polarity, intermolecular interactions and the ability of the solvents to form a hydrogen bond with the solute molecules [20].

### 2.4. Thermodynamic Properties for the Solution

For a real solution, it is important to study the dissolution behaviors of solutes in different solvents. Enthalpy change$\;(\Delta H_{d}$) reflects the change in the energy of the dissolution process, which is closely related to the interactions of solute–solute, solvent–solvent and solvent–solute, while entropy change ($\Delta S_{d}$) reflects the information of the disorder degree of a system.

The relationship between the logarithm of the molar fraction solubility of thiamine nitrate and the reciprocal of the absolute temperature in different binary mixtures are shown in Figure 7, Figure 8 and Figure 9. Observing the van’t Hoff model shown in Equation (4), it can be deduced that the molar enthalpy of dissolution $\Delta H_{d}$ and the molar entropy of dissolution $\Delta S_{d}$ can be obtained from the slope and the intercept of the fitting line, respectively. Additionally, the results are given in Table 4. To reduce the error in the calculation, we calculate the average temperature${\;T}_{\mathrm{mean}}$, which is defined in Equation (5) [21]:(5)$$T_{mean}=N/\sum_{i}^{N}\left(1/T\right)$$
where *N* is the number of experimental points. In this study, $T_{mean}$ = 295.21 K is used to calculate thermodynamic functions. In addition, the molar Gibbs energy of dissolution can be calculated using the Gibbs–Helmholtz equation, as evidenced by Equation (6):(6)$$\Delta G_{d}=\Delta H_{d}{\;-\;T}_{mean}\Delta S_{d}$$

To compare the relative contributions of enthalpy and entropy to the Gibbs energy change, *ζ_H_* and *ζ_TS_*, which represent the contributions of enthalpy and entropy, respectively, were defined as follows:(7)$${\;\zeta }_{H}=\left|\;\Delta H_{d}\;\right|/\left(\left|\;\Delta H_{d}\;\right|+\left|{\;T}_{mean}\Delta S_{d}\;\right|\right)$$
(8)$${\;\zeta }_{TS}=\left|{\;T}_{\mathrm{mean}}\Delta S_{d}\;\right|/\left(\left|\Delta H_{d}\;\right|+\left|{\;T}_{\mathrm{mean}}\Delta S_{d\;}\right|\right)$$

The obtained thermodynamic parameters of thiamine nitrate in the dissolving process are given in Table 4. It was worth noting that the molar enthalpy energy change ($\Delta H_{d}$) of thiamine nitrate in all cases was positive, from an energetic aspect, meaning that the dissolution processes were endothermic [22], which was consistent with the fact that the solubility of thiamine nitrate increased with increasing temperature in all three types of binary solvent mixtures. This phenomenon is explained by the fact that interaction forces between the solute and solvent are weaker than those corresponding to binary solvents themselves, hence, more energy is needed to overcome the cohesive force of the solute and the solvent in the dissolution process [23]. In addition, from the values of ${\;\zeta }_{H}$ and ${\;\zeta }_{TS}$ in Table 4, it is apparent that the values of ${\;\zeta }_{TS}$ were smaller than the values of ${\;\zeta }_{H}$ in all cases. Herein, we can draw a conclusion that the main contributor to the molar Gibbs energy change was entropy in all situations. In brief, these results are helpful for the optimization of the dissolution and crystallization processes of thiamine nitrate.

## 3. Experimental

### 3.1. Materials

Thiamine nitrate was supplied by Xinfa Pharmaceutical Co., Ltd. (Shandong, China). The organic solvents were provided by Tianjin Jiangtian Chemical Technique Co., Ltd. (Tianjin, China). Distilled–deionized water with a resistivity of 18.2 Ωm was used throughout and was made in our laboratory using the NANOPURE system from BARNSTEAD (Thermo Scientific Co., Shanghai, China). All chemicals were used without further purification. More detailed information about the materials used in this work have been listed in Table 5.

### 3.2. Solubility Measurements

In this work, the laser monitoring technique was used to determine the solubility of thiamine nitrate. The apparatus was similar to that described in the literature [24,25]. The method is based on the Lambert−Beer law, which correlates light intensity and concentration of particles in suspension. All experiments were performed in a 100-mL jacketed crystallizer with a mechanical agitation applied as a dissolver. A thermometer with an uncertainty of ±0.1 K inside the vessel displayed the real temperature. The circulating water bath (CHY1015, Shanghai Shunyu Hengping Scientific Instrument Co., Ltd., Shanghai, China) was applied to control the temperature within the error range of ±0.1 K. The transmitted laser (JD-3, Department of Peking University, Beijing, China) was employed to monitor the dissolution by the change in the intensity of the solution. The solute and solvents were weighted via an electronic analytical balance (AL204, Meteler Toledo, Greifensee, Switzerland) with an accuracy of ±0.0001 g.

At a fixed temperature, predetermined masses of solvents (50 g) were added to the jacketed crystallizer. A condenser was employed to prevent the evaporation of the solvents. A fixed amount of thiamine nitrate was added to the vessel when the temperature was stable. At first, the laser beam was blocked by the undissolved particles in the solution, so the intensity of the laser beam through the crystallizer was low. With the dissolution of the solute, the intensity increases gradually. When the thiamine nitrate had just disappeared, the intensity reached the maximum. Then, an additional solute of known mass (0.1–0.5 mg) was added to the crystallizer, and subsequently the intensity of the laser decreased immediately. The intensity of the laser increased gradually along with the dissolution of the particles and reached the former constant. This process was repeated until the particles could not dissolve and the laser intensity could be kept constant.

The mixture was considered as reaching phase equilibrium. Then, the total consumption of the solute was recorded. Each point was repeated at least three times. The solubility of thiamine nitrate described in molar fraction $x_{3}$ in different binary solvent mixtures was calculated using Equation (9) and the composition of the solvent mixtures was expressed using Equation (10), as follows [26]:(9)$$x_{3}=\frac{m_{3}/M_{3}}{m_{1}/M_{1}{+m}_{2}/M_{2}{+m}_{3}/M_{3}}$$
(10)$$x_{1}^{0}=\frac{m_{1}/M_{1}}{m_{1}/M_{1}+m_{2}/M_{2}}\;$$
where *m*_3_, *m*_1_ and *m*_2_ represent the mass of thiamine nitrate, organic solvents (methanol, acetone, isopropanol) and water, respectively. Similarly, *M*_3_, *M*_1_ and *M*_2_ refer to the molar mass of thiamine nitrate, organic solvents (methanol, acetone, isopropanol) and water, respectively.

### 3.3. X-ray Powder Diffraction

To ensure that the crystal form of thiamine nitrate remains the same during the experiments, powder X-ray diffraction (PXRD) patterns of suspension of thiamine nitrate in different solvent mixtures and temperatures agitated for more than 24 h were measured at 40 kV and 100 mA, respectively. Additionally, the data collection was obtained via Rigaku D/max-2500 (Rigaku, Tokyo, Japan) using Cu Kα radiation (1.5405 Å) in the 2-theta range of 2° to 50° and at a scanning rate of 1 step/s.

### 3.4. Characterization via TGA/DSC

Thermogravimetric Analysis/Differential Scanning Calorimetry (TGA/DSC) (Model TGA/DSC, Mettler-Toledo, Greifensee, Switzerland) can simultaneously measure the properties of the sample, such as heat flow, transition temperature and weight change. In this experiment, it was employed to ensure the thermal analysis of thiamine nitrate. The measurements were carried out under the protection of nitrogen, with a heating rate of 5 K/min. The amount of the sample used was about 5–10 mg.

## 4. Thermodynamic Models

### 4.1. Modified Apelblat Equation

The modified Apelblat equation is a widely used semi-empirical equation that was previously put forward by Apelblat [27]. It is a quite accurate mathematical description for binary solid–liquid phase equilibrium. Due to its simplicity, it is widely used for the prediction of solid–liquid equilibrium data. Its simplified form is shown as Equation (11):(11)$${\ln\;x}_{3}=A+\frac{B}{T/K}+C\;\ln\;(T/K)$$
where${\;x}_{3}$ is the molar fraction solubility of solute; *T* is the absolute experimental temperature and *A*, *B* and *C* are empirical parameters. *A* and *B* reflect the non-idealities of the real solution in terms of the variation of activity coefficients in the solution, while *C* represents the effect of temperature on the fusion enthalpy [28,29].

### 4.2. λh Model

The *λh* model equation which was originally developed by Buchowski et al. [30] expresses the non-ideality and the enthalpy of the solution. *λ* and *h* are the two parameters of the model. It can be shown as Equation (12).
(12)$$\ln\left(1+\lambda \;\frac{1-x}{x_{3}}\right)=\lambda h\left(\frac{1}{T}-\frac{1}{T_{m}}\right)$$
where $x_{3}$ is the molar fraction solubility of the solute, $T_{m}\;$ is the melting temperature of the solute abd *T* stands for the absolute temperature. The parameters *λ* and *h* are determined by the correlation of the solubility data.

### 4.3. Van’t Hoff Equation

For a real solution, the standard van’t Hoff equation [25] expresses a linear relationship between the logarithm of the molar fraction solubility and the reciprocal of the absolute temperature, and it can be described as Equation (13) [31].
(13)$$\ln\;x_{3}=\frac{\Delta S_{d}}{R}-\frac{\Delta H_{d}}{RT}$$
where $x_{3}$ is the molar fraction solubility of the solute; $\Delta H_{d}$ and $\Delta S_{d}$ are the dissolution enthalpy and entropy, respectively; *T* is the absolute temperature of the solution and *R* is the gas constant.

### 4.4. The Local Composition Model: NRTL Model

The NRTL (Non-Random Two-Liquid) model is proposed on the basis of the local composition concept. It is used extensively in describing vapor–liquid, liquid–liquid and liquid–solid phase equilibriums [32]. According to the solid–liquid phase equilibrium theory and the solute–solvent interactions, the local composition equation can be simplified and expressed by Equation (14):(14)$${\ln\;x}_{i}=\frac{\Delta_{fus}H}{R}\left(\frac{1}{T_{m}}-\frac{1}{T}\right)-\ln\;\gamma_{i}$$
where $\Delta_{fus}H$ and $T_{m}$ stand for the enthalpy of fusion and melting temperature of solute. The activity coefficient$\;\gamma_{i}$ expressed by the NRTL model is presented as Equation (15).
(15)$$\begin{array}{cc} ln\;\gamma_{i}= & \left(G_{ji}x_{j}+G_{kj}x_{k}\right)\left(\tau_{ji}G_{ji}x_{j}+\tau_{ki}G_{ki}x_{k}\right)/{\left(x_{i}+x_{j}G_{j}+x_{k}G_{ki}\right)}^{2}\\ & +\left[\tau_{ij}G_{ij}x_{j}^{2}+G_{ij}G_{kj}x_{j}x_{k}\left(\tau_{ij}-\tau_{kj}\right)\right]/{\left(x_{j}+x_{i}G_{ij}+x_{k}G_{kj}\right)}^{2}\\ & +\left[\tau_{ik}G_{ij}x_{j}^{2}+G_{ij}G_{kj}x_{j}x_{k}\left(\tau_{ik}-\tau_{jk}\right)\right]/{\left(x_{k}+x_{i}G_{ik}+x_{j}G_{jk}\right)}^{2} \end{array}$$
where$\;G_{ij}$, $G_{ik}$, $G_{ji}$, $G_{jk}$, $G_{ki}$, $G_{kj}$, $\tau_{ij}$, $\tau_{ik}$, $\tau_{ji}$, $\tau_{jk}$, $\tau_{ki}$ and $\tau_{kj}$ are parameters of this model. The definition of these terms can be expressed as Equations (16) and (17), respectively:(16)$$G_{ij}=exp\left(-\alpha_{ij}\tau_{ij}\right)$$
(17)$$\tau_{ij}=\left(g_{ij}-g_{jj}\right)/RT=\Delta g_{ij}/RT$$
where $\Delta g_{ij}$ represents the Gibbs energy of the intermolecular interactions which are independent of the compositions and temperatures. $\alpha_{ij}$ is an adjustable empirical constant between 0 and 1 and is a criterion of the non-randomness of the solution.

In this work, the four models were used to correlate the experimental solubility using the MATLAB program, and the calculated solubility is listed in Table 1, Table 2 and Table 3. The parameters of the four models were obtained via the least squares method, and they are given in Table 4, Table 6, Table 7 and Table 8. To test the applicability and accuracy of the models used in this paper, the average relative deviation (*ARD*%) was defined as Equation (18), which is also presented in Table 4, Table 6, Table 7 and Table 8 to assess the accuracy of different models in this paper.
(18)$$\;ARD\%=\frac{100}{N}\overset{N}{\underset{i=1}{\sum }}\left|\;\frac{x_{3,i}^{exp}-{\;x}_{3,i}^{cal}}{x_{3,i}^{exp}}\;\right|$$
where *N* is equal to the number of experimental points, $x_{3,i}^{exp}\;$ and $x_{3,i}^{cal}\;$ refer to the experimental and calculated solubility, respectively.

From the *ARD*% which is presented in Table 4, Table 6, Table 7 and Table 8, it could be concluded that all the four models could give satisfactory correlation results, especially the modified Apelblat model.

**Table 1 molecules-28-05012-t001:** Molar fraction solubility ($x_{3}$) of thiamine nitrate in the binary methanol (1) + water (2) solvent mixtures at different temperatures ranging from 278.15 to 313.15 K (*p* = 0.1 MPa) ^abcde^.

$x_{1}^{0}$	$10^{4}x_{3}^{exp}\;$	$10^{4}x_{3}^{cal}$ (Equation (11) Apelbat)	$10^{4}x_{3}^{cal}$ (Equation (12) *λh*)	$10^{4}x_{3}^{cal}$ (Equation (13) Van’t Hoff)	$10^{4}x_{3}^{cal}$ (Equation (14) NRTL)
	*T* = 278.15 K				
0.10	7.20	7.11	6.67	6.65	6.28
0.20	7.61	7.52	6.87	6.85	7.28
0.30	8.02	7.98	7.31	7.29	7.54
0.40	7.50	7.43	6.94	6.92	7.08
0.50	6.82	6.84	6.18	6.16	6.09
0.60	5.51	5.50	4.77	4.76	4.83
0.70	4.13	4.14	3.60	3.59	3.55
0.80	2.89	2.86	2.45	2.44	2.43
0.90	1.85	1.73	1.49	1.49	1.55
	*T* = 283.15 K				
0.10	8.65	8.59	8.35	8.34	7.93
0.20	9.20	9.07	8.71	8.70	9.12
0.30	9.71	9.59	9.23	9.22	9.41
0.40	9.04	8.96	8.70	8.69	8.82
0.50	8.13	8.05	7.70	7.69	7.60
0.60	6.52	6.38	6.00	5.99	6.04
0.70	4.84	4.80	4.52	4.51	4.47
0.80	3.33	3.31	3.09	3.09	3.08
0.90	2.07	2.07	1.93	1.93	1.98
	*T* = 288.15 K				
0.10	10.26	10.41	10.39	10.38	9.95
0.20	10.87	11.01	10.95	10.95	11.35
0.30	11.49	11.61	11.56	11.55	11.66
0.40	10.75	10.85	10.82	10.82	10.92
0.50	9.56	9.56	9.53	9.52	9.41
0.60	7.48	7.53	7.48	7.48	7.51
0.70	5.65	5.66	5.63	5.63	5.58
0.80	3.85	3.91	3.88	3.88	3.86
0.90	2.56	2.50	2.47	2.47	2.50
	*T* = 293.15 K				
0.10	12.58	12.66	12.82	12.82	12.41
0.20	13.22	13.46	13.67	13.68	14.04
0.30	13.97	14.14	14.37	14.38	14.37
0.40	13.00	13.19	13.36	13.37	13.44
0.50	11.37	11.46	11.71	11.71	11.59
0.60	8.78	9.01	9.26	9.26	9.27
0.70	6.77	6.78	6.97	6.97	6.92
0.80	4.67	4.69	4.83	4.83	4.82
0.90	3.29	3.07	3.13	3.13	3.15
	*T* = 298.15 K				
0.10	15.42	15.43	15.72	15.73	15.39
0.20	16.52	16.53	16.95	16.96	17.28
0.30	17.36	17.31	17.75	17.76	17.61
0.40	16.10	16.07	16.39	16.40	16.44
0.50	13.80	13.86	14.30	14.31	14.19
0.60	10.93	10.93	11.39	11.39	11.39
0.70	8.16	8.22	8.56	8.57	8.53
0.80	5.72	5.71	5.97	5.97	5.99
0.90	3.97	3.81	3.95	3.95	3.94
	*T* = 303.15 K				
0.10	18.95	18.85	19.15	19.16	18.96
0.20	20.60	20.41	20.86	20.87	21.14
0.30	21.39	21.30	21.76	21.78	21.46
0.40	19.82	19.64	19.98	19.99	20.01
0.50	17.03	16.90	17.34	17.36	17.29
0.60	13.44	13.42	13.91	13.92	13.91
0.70	10.09	10.09	10.45	10.46	10.47
0.80	7.07	7.06	7.33	7.34	7.39
0.90	4.87	4.77	4.94	4.94	4.90
	*T* = 308.15 K				
0.10	23.17	23.06	23.19	23.19	23.25
0.20	25.41	25.32	25.51	25.52	25.74
0.30	26.39	26.33	26.52	26.53	26.02
0.40	24.05	24.07	24.20	24.21	24.23
0.50	20.73	20.74	20.92	20.93	20.95
0.60	16.90	16.69	16.89	16.90	16.90
0.70	12.60	12.53	12.68	12.68	12.77
0.80	8.86	8.83	8.95	8.95	9.07
0.90	6.19	6.05	6.13	6.14	6.06
	*T* = 313.15 K				
0.10	28.19	28.27	27.92	27.90	28.34
0.20	31.45	31.54	31.02	31.01	31.18
0.30	32.62	32.68	32.14	32.12	31.41
0.40	29.51	29.55	29.15	29.14	29.21
0.50	25.62	25.63	25.09	25.08	25.27
0.60	20.86	20.97	20.39	20.39	20.43
0.70	15.69	15.72	15.29	15.29	15.50
0.80	11.16	11.19	10.86	10.86	11.07
0.90	7.66	7.75	7.57	7.57	7.45

^a^$\;x_{1}^{0}$ is the initial molar fraction of methanol in the binary solvent mixture;${\;x}_{3}^{exp}$ is the experimentally determined solubility; $x_{3}^{cal}$ (Equation (11)), $x_{3}^{cal}\;$ (Equation (12)), $x_{3}^{cal}$ (Equation (13)) and $x_{3}^{cal}$ (Equation (14)) are the calculated solubilities obtained using Equations (11)–(14), respectively. ^b^ The standard uncertainty of temperature is u_c_ (*T*) = 0.1 K. ^c^ The relative standard uncertainty of the solubility measurement is u_r_ ($x_{3}^{exp}$) = 0.07. ^d^ The relative uncertainty of pressure is u_r_ (*p*) = 0.05. ^e^ The relative standard uncertainty in the molar fraction of methanol (1) in the solvent mixtures is u_r_ ($x_{1}^{0}$) = 0.005.

**Table 2 molecules-28-05012-t002:** Molar fraction solubility (*x*_3_) of thiamine nitrate in the binary acetone (1) + water (2) solvent mixtures at different temperatures ranging from 278.15 to 313.15 K (*p* = 0.1 MPa) ^abcde^.

$x_{1}^{0}$	$10^{4}x_{3}^{exp}\;$	$10^{4}x_{3}^{cal}$ (Equation (11) Apelbat)	$10^{4}x_{3}^{cal}$ (Equation (12) *λh*)	$10^{4}x_{3}^{cal}$ (Equation (13) Van’t Hoff)	$10^{4}x_{3}^{cal}$ (Equation (14) NRTL)
	*T* = 278.15 K				
0.10	12.70	12.65	12.17	12.13	12.21
0.20	12.53	12.35	12.00	11.95	12.37
0.30	9.97	10.03	9.45	9.41	9.67
0.40	6.78	6.91	6.21	6.18	6.42
0.50	4.60	4.57	4.40	4.37	3.78
0.60	2.18	2.17	2.17	2.16	1.98
0.70	0.89	0.87	0.83	0.82	0.88
	*T* = 283.15 K				
0.10	14.94	15.01	14.78	14.75	14.74
0.20	14.52	14.67	14.50	14.48	14.77
0.30	11.52	11.57	11.31	11.29	11.50
0.40	7.81	7.81	7.49	7.47	7.63
0.50	5.15	5.14	5.08	5.06	4.49
0.60	2.55	2.52	2.52	2.51	2.35
0.70	1.02	1.02	0.99	0.99	1.04
	*T* = 288.15 K				
0.10	17.54	17.82	17.83	17.83	17.73
0.20	17.36	17.41	17.42	17.41	17.56
0.30	13.51	13.43	13.47	13.46	13.63
0.40	9.12	8.95	8.98	8.97	9.04
0.50	5.81	5.81	5.83	5.83	5.32
0.60	2.92	2.91	2.91	2.91	2.78
0.70	1.17	1.19	1.19	1.19	1.23
	*T* = 293.15 K				
0.10	21.54	21.18	21.39	21.40	21.26
0.20	20.61	20.64	20.80	20.81	20.81
0.30	15.76	15.67	15.94	15.95	16.10
0.40	10.45	10.39	10.70	10.71	10.67
0.50	6.50	6.58	6.67	6.68	6.28
0.60	3.25	3.35	3.35	3.36	3.27
0.70	1.34	1.39	1.41	1.41	1.45
	*T* = 298.15 K				
0.10	25.22	25.17	25.50	25.53	25.38
0.20	24.31	24.46	24.70	24.73	24.58
0.30	18.47	18.38	18.78	18.80	18.94
0.40	12.20	12.21	12.68	12.70	12.54
0.50	7.47	7.49	7.61	7.62	7.38
0.60	3.84	3.84	3.84	3.85	3.85
0.70	1.67	1.64	1.67	1.67	1.71
	*T* = 303.15 K				
0.10	29.99	29.94	30.25	30.28	30.19
0.20	29.24	28.96	29.19	29.22	28.94
0.30	21.60	21.64	22.01	22.04	22.22
0.40	14.45	14.51	14.96	14.98	14.70
0.50	8.58	8.54	8.64	8.66	8.64
0.60	4.43	4.39	4.39	4.40	4.50
0.70	1.95	1.93	1.96	1.96	2.00
	*T* = 308.15 K				
0.10	35.30	35.61	35.71	35.73	35.80
0.20	34.23	34.25	34.32	34.34	33.96
0.30	25.34	25.58	25.69	25.70	25.98
0.40	17.27	17.42	17.57	17.57	17.17
0.50	9.86	9.77	9.79	9.80	10.09
0.60	5.06	5.01	5.00	5.01	5.26
0.70	2.27	2.29	2.30	2.30	2.34
	*T* = 313.15 K				
0.10	42.51	42.36	41.96	41.92	42.33
0.20	40.42	40.48	40.18	40.14	39.74
0.30	30.50	30.36	29.86	29.83	30.29
0.40	21.23	21.13	20.53	20.51	20.01
0.50	11.15	11.21	11.06	11.04	11.75
0.60	5.64	5.69	5.68	5.67	6.12
0.70	2.72	2.72	2.68	2.68	2.72

^a^$\;x_{1}^{0}$ is the initial molar fraction of acetone in the binary solvent mixture;${\;x}_{3}^{exp}$ is the experimentally determined solubility; $x_{3}^{cal}$ (Equation (11)), $x_{3}^{cal}$ (Equation (12)), $x_{3}^{cal}$ (Equation (13)) and $x_{3}^{cal}$ (Equation (14)) are the calculated solubilities obtained using Equations (11)–(14), respectively. ^b^ The standard uncertainty of temperature is u_c_ (*T*) = 0.1 K. ^c^ The relative standard uncertainty of the solubility measurement is u_r_ ($x_{3}^{exp}$) = 0.07. ^d^ The relative uncertainty of pressure is u_r_ (*p*) = 0.05. ^e^ The relative standard uncertainty in molar fraction of acetone (1) in the solvent mixtures is u_r_ ($x_{1}^{0}$) = 0.005.

**Table 3 molecules-28-05012-t003:** Molar fraction solubility ($x_{3}$) of thiamine nitrate in the binary isopropanol (1) + water (2) solvent mixtures at different temperatures ranging from 278.15 to 313.15 K (*p* = 0.1 MPa) ^abcde^.

$x_{1}^{0}$	$10^{4}x_{3}^{exp}\;$	$10^{4}x_{3}^{cal}$ (Equation (11) Apelbat)	$10^{4}x_{3}^{cal}$ (Equation (12) *λh*)	$10^{4}x_{3}^{cal}$ (Equation (13) Van’t Hoff)	$10^{4}x_{3}^{cal}$ (Equation (14) NRTL)
	*T* = 278.15 K				
0.10	8.13	8.08	7.89	7.89	8.47
0.20	7.89	7.68	7.39	7.39	7.49
0.30	6.00	5.96	5.42	5.42	5.81
0.40	3.89	3.93	3.63	3.63	3.99
0.50	2.73	2.65	2.28	2.28	2.43
0.60	1.24	1.35	1.19	1.19	1.23
0.70	0.50	0.52	0.44	0.44	0.48
	*T* = 283.15 K				
0.10	10.10	10.18	10.07	10.07	10.44
0.20	9.41	9.47	9.31	9.31	9.20
0.30	7.21	7.19	6.89	6.89	7.13
0.40	4.69	4.79	4.62	4.62	4.90
0.50	3.07	3.11	2.90	2.90	2.98
0.60	1.58	1.57	1.48	1.48	1.52
0.70	0.60	0.61	0.56	0.56	0.60
	*T* = 288.15 K				
0.10	12.63	12.76	12.74	12.74	12.87
0.20	11.39	11.66	11.63	11.63	11.30
0.30	8.65	8.73	8.68	8.68	8.75
0.40	5.91	5.87	5.84	5.84	6.03
0.50	3.65	3.70	3.66	3.66	3.65
0.60	1.98	1.85	1.84	1.84	1.87
0.70	0.74	0.72	0.71	0.71	0.74
	*T* = 293.15 K				
0.10	16.15	15.93	15.99	15.99	15.93
0.20	14.28	14.33	14.43	14.43	13.96
0.30	10.63	10.67	10.84	10.84	10.76
0.40	7.55	7.22	7.31	7.31	7.44
0.50	4.45	4.47	4.59	4.59	4.48
0.60	2.27	2.21	2.27	2.27	2.29
0.70	0.87	0.87	0.90	0.90	0.91
	*T* = 298.15 K				
0.10	19.85	19.80	19.92	19.92	19.65
0.20	17.68	17.57	17.77	17.77	17.24
0.30	13.09	13.11	13.45	13.45	13.25
0.40	8.73	8.90	9.09	9.09	9.07
0.50	5.43	5.48	5.71	5.71	5.50
0.60	2.64	2.67	2.78	2.78	2.80
0.70	1.08	1.08	1.13	1.13	1.13
	*T* = 303.15 K				
0.10	24.43	24.50	24.63	24.63	24.26
0.20	21.76	21.52	21.72	21.72	21.30
0.30	16.23	16.19	16.56	16.56	16.34
0.40	10.83	11.02	11.22	11.22	11.15
0.50	6.87	6.80	7.05	7.05	6.76
0.60	3.22	3.26	3.38	3.38	3.43
0.70	1.35	1.34	1.40	1.40	1.39
	*T* = 308.15 K				
0.10	30.10	30.19	30.25	30.25	30.05
0.20	26.13	26.31	26.39	26.39	26.22
0.30	20.16	20.10	20.26	20.26	20.21
0.40	13.79	13.67	13.76	13.76	13.79
0.50	8.58	8.54	8.65	8.65	8.31
0.60	3.93	4.03	4.08	4.08	4.20
0.70	1.67	1.70	1.73	1.73	1.70
	*T* = 313.15 K				
0.10	37.11	37.06	36.90	36.90	37.34
0.20	32.12	32.10	31.86	31.86	32.54
0.30	25.01	25.05	24.63	24.63	25.06
0.40	16.99	17.00	16.76	16.76	16.99
0.50	10.79	10.83	10.54	10.54	10.23
0.60	5.10	5.03	4.89	4.89	5.15
0.70	2.19	2.18	2.12	2.12	2.10

^a^$\;x_{1}^{0}$ is the initial molar fraction of isopropanol in the binary solvent mixture;${\;x}_{3}^{exp}$ is the experimentally determined solubility; $x_{3}^{cal}$ (Equation (11)), $x_{3}^{cal}$ (Equation (12)), $x_{3}^{cal}$ (Equation (13)) and $x_{3}^{cal}$ (Equation (14)), are the calculated solubilities obtained using Equations (11)–(14), respectively. ^b^ The standard uncertainty of temperature is u_c_ (*T*) = 0.1 K. ^c^ The relative standard uncertainty of the solubility measurement is u_r_ ($x_{3}^{exp}$) = 0.07. ^d^ The relative uncertainty of pressure is u_r_ (*p*) = 0.05. ^e^ The relative standard uncertainty in molar fraction of isopropanol (1) in the solvent mixtures is u_r_ ($x_{1}^{0}$) = 0.005.

**Table 4 molecules-28-05012-t004:** Parameters of the van’t Hoff equation and dissolution thermodynamic properties for thiamine nitrate in different binary solvent mixtures.

$x_{1}^{0}\;$	Δ*H_d_* kJ·mol^−1^	Δ*G_d_* kJ·mol^−1^	Δ*S_d_* J·mol^−1^	*T*Δ*S_d_* kJ·mol^−1^	*ζ_H_*	*ζ_TS_*	10^2^ *ARD*
Methanol + Water							
0.10	29.6733	16.1355	45.8583	13.5378	0.6867	0.3133	2.34
0.20	31.2393	15.9664	51.7357	15.2729	0.6716	0.3284	3.18
0.30	30.6851	15.8478	50.2601	14.8373	0.6741	0.3259	2.96
0.40	29.7402	16.0330	46.4320	13.7072	0.6845	0.3155	2.47
0.50	29.0576	16.3619	43.0056	12.6957	0.6959	0.3041	3.39
0.60	30.1090	16.9308	44.6402	13.1782	0.6956	0.3044	4.69
0.70	29.9773	17.6300	41.8254	12.3473	0.7083	0.2917	3.18
0.80	30.9026	18.5232	41.9340	12.3793	0.7140	0.2860	4.90
0.90	33.6244	19.5653	47.6240	14.0591	0.7052	0.2948	5.48
							overall
						10^2^ *ARD* = 3.62	
Acetone + Water							
0.10	25.6630	14.9068	36.4359	10.7563	0.7047	0.2953	1.61
0.20	25.0627	14.9789	34.1580	10.0838	0.7131	0.2869	1.12
0.30	23.8790	15.6398	27.9095	8.2392	0.7435	0.2565	2.09
0.40	24.8188	16.6124	27.7983	8.2063	0.7515	0.2485	3.76
0.50	19.1458	17.8090	4.5285	1.3369	0.9347	0.0653	1.79
0.60	20.0210	19.4927	1.7895	0.5283	0.9743	0.0257	1.12
0.70	24.4079	21.5880	9.5521	2.8199	0.8964	0.1036	2.49
							overall
						10^2^ *ARD* = 2.00	
Isopropanol + Water							
0.10	31.9093	15.5775	55.32	16.3318	0.6615	0.3385	0.90
0.20	30.2313	15.8419	48.74	14.3893	0.6775	0.3225	1.62
0.30	31.3064	16.5358	50.03	14.7705	0.6794	0.3206	2.91
0.40	31.6377	17.4994	47.89	14.1383	0.6911	0.3089	2.73
0.50	31.7249	18.6421	44.32	13.0827	0.7080	0.2920	4.56
0.60	29.3288	20.3879	30.29	8.9409	0.7664	0.2336	4.40
0.70	32.6326	22.6357	33.86	9.9969	0.7655	0.2345	5.27
							overall
						10^2^ *ARD* = 3.20	

**Table 5 molecules-28-05012-t005:** Sources and mass fraction purity of materials used in the experiment.

Chemical Name	Source	Mass Fraction Purity	Molar Mass (g mol^−1^)	Purification Method	Analysis Method
Thiamine nitrate	Xinfa Pharmaceutical Co., Ltd., Shandong, China	0.990	327.36	none	HPLC ^a^
Methanol	Tianjin Kewei Chemical Reagent Co., Tianjin, China	0.995	32.04	none	GC ^b^
Acetone	Tianjin Kewei Chemical Reagent Co., Tianjin, China	0.995	58.08	none	GC ^b^
Isopropanol	Tianjin Kewei Chemical Reagent Co., Tianjin, China	0.995	60.07	none	GC ^b^

^a^ High-performance liquid chromatography; ^b^ gas chromatography; both the analysis method and the mass fraction purity were provided by the suppliers.

**Table 6 molecules-28-05012-t006:** Parameters of the modified Apelblat equation for thiamine nitrate in different binary solvent mixtures ranging from 278.15 to 313.15 K (*p* = 0.1 MPa).

$x_{1}^{0}$	A	B	C	10^2^ *ARD*
Methanol + Water				
0.10	−205.4794	5849.7516	31.4844	0.70
0.20	−281.2900	9084.9199	42.8977	0.92
0.30	−275.3669	8875.7786	41.9894	0.62
0.40	−220.7492	6526.4412	33.7732	0.68
0.50	−335.5374	11,708.2293	50.8437	0.45
0.60	−456.0720	16,977.9187	68.8551	0.95
0.70	−450.6844	16,735.6338	68.0021	0.34
0.80	−493.9870	18,566.2180	74.4608	0.59
0.90	−439.1955	15,851.2381	66.3712	2.59
				overall
				10^2^ *ARD* = 0.87
Acetone + Water				
0.10	−65.1099	−633.7365	10.6929	0.55
0.20	−111.1695	2121.8809	17.2071	0.58
0.30	−229.7801	7508.6759	34.8036	0.53
0.40	−395.9708	14,799.3341	59.6085	0.79
0.50	−175.3088	5513.4288	26.2601	0.53
0.60	−27.3943	−1180.3574	4.1226	1.00
0.70	−214.5103	6670.7406	32.1921	1.29
				overall
				10^2^ *ARD* = 0.75
Isopropanol + Water				
0.10	−64.2894	−668.1882	10.5844	0.59
0.20	−115.2615	1772.2994	18.0731	1.05
0.30	−284.6482	9217.4086	43.3684	0.40
0.40	−234.8014	6940.1326	35.8924	1.57
0.50	−462.8546	17,098.5253	69.8534	1.08
0.60	−423.9030	15,543.7370	63.8059	3.15
0.70	−510.4195	19,059.8807	76.7614	1.22
				overall
				10^2^ *ARD* = 1.29

**Table 7 molecules-28-05012-t007:** Parameters of the *λh* equation for thiamine nitrate in different binary solvent mixtures ranging from 278.15 to 313.15 K (*p* = 0.1 MPa).

$x_{1}^{0}$	*λ*	*h*	10^2^ *ARD*
Methanol + Water			
0.10	0.1536	23,017.5768	2.26
0.20	0.2140	17,429.4069	3.12
0.30	0.2048	17,875.9748	2.90
040	0.1620	21,874.8842	2.40
0.50	0.1260	27,437.3221	3.30
0.60	0.1190	30,144.8550	4.62
0.70	0.0874	40,849.2903	4.28
0.80	0.0708	52,042.5106	4.83
0.90	0.0727	55,293.3980	5.44
			overall
			10^2^ *ARD* = 3.68
Acetone + Water			
0.10	0.1284	23,604.4564	1.52
0.20	0.1122	26,311.1643	1.03
0.30	0.0693	40,376.2910	1.94
0.40	0.0547	53,256.0104	3.61
0.50	0.0119	182,114.7271	1.59
0.60	0.0071	322,587.8581	1.03
0.70	0.0067	428,347.4676	2.39
			overall
			10^2^ *ARD* = 1.87
Isopropanol + Water			
0.10	0.2814	13,561.1004	0.80
0.20	0.1905	18,929.6668	1.48
0.30	0.1712	21,826.9691	2.85
0.40	0.1219	30,987.5052	3.43
0.50	0.0773	48,938.2647	4.50
0.60	0.0254	137,357.2310	6.07
0.70	0.0176	221,162.6311	8.12
			overall
			10^2^ *ARD* = 3.89

**Table 8 molecules-28-05012-t008:** Parameters of the NRTL equation for thiamine nitrate in different binary solvent mixtures ranging from 278.15 to 313.15 K (*p* = 0.1 MPa).

Parameters	Methanol + Water	Acetone + Water	Isopropanol + Water
10^−4^∆g12	2.4821	4.4086	0.2994
10^−4^∆g13	1.8502	2.6368	−1.6948
10^−4^∆g21	−0.6585	0.2024	2.4184
10^−4^∆g23	−0.2432	−0.4181	0.8077
10^−4^∆g31	−0.8793	−1.5136	2.7812
10^−4^∆g32	−0.2137	0.5935	0.5740
10^2^ *ARD*	3.62	3.03	2.48

## 5. Conclusions

In this study, the solubility was experimentally measured over temperatures ranging from 278.15 to 313.15 K under atmospheric pressure using the dynamic method. In general, the dissolving capacity rankings of thiamine nitrate in the binary solvent mixtures at a constant temperature were methanol + water > acetone + water > isopropanol + water at the low ratio of water, which is in conformity with the empirical rule “like dissolves like”. Interestingly, in the high ratio of water systems, the dissolving capacity rankings were water + acetone > water + methanol > water + isopropanol. The occurrence of these phenomena has a complex thermodynamic basis, such as the mechanism of cosolvency, with “cosolvency” meaning that the solubility of a substance in a certain proportion of the mixed solvent is greater than that of any single solvent.

The experimental solubility of thiamine nitrate in solvent mixtures was correlated based on the modified Apelblat model, *λh* model, van’t Hoff model and NRTL model, respectively. It turns out that all the selected thermodynamic models could give satisfactory correlation results. In addition, the thermodynamic parameters of the dissolution process for this system were obtained using the van’t Hoff model, which indicated that the dissolution of the thiamine nitrate in the selected solvents was endothermic. On the basis of the above results, the experimental solubility data and equations presented in this study can be used to optimize the practical crystallization conditions of thiamine nitrate.

## Figures and Tables

**Figure 1 molecules-28-05012-f001:**
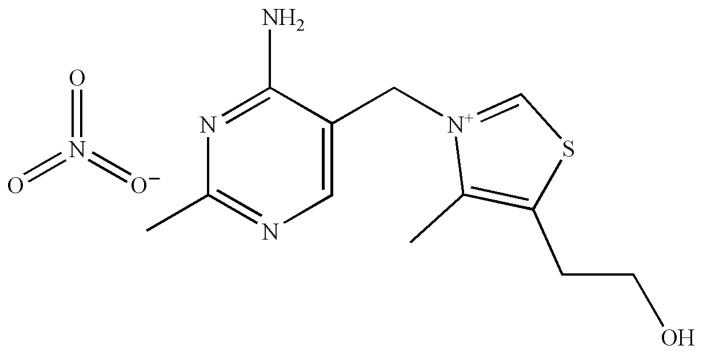
Chemical structure of thiamine nitrate.

**Figure 2 molecules-28-05012-f002:**
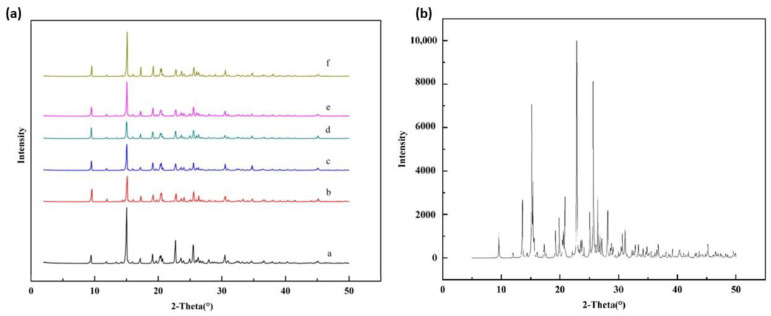
(**a**) Powder X-ray diffraction (PXRD) patterns for the excess solid of thiamine nitrate under different conditions: (a = raw materials; b, c, d, e, f = sediments in binary solvents methanol ($x_{1}^{0}$ = 0.10, 0.30, 0.50, 0.70, 0.90) + water at *T* = 298.15 K, respectively). (**b**) The simulated XRD pattern sourced from the single-crystal XRD.

**Figure 3 molecules-28-05012-f003:**
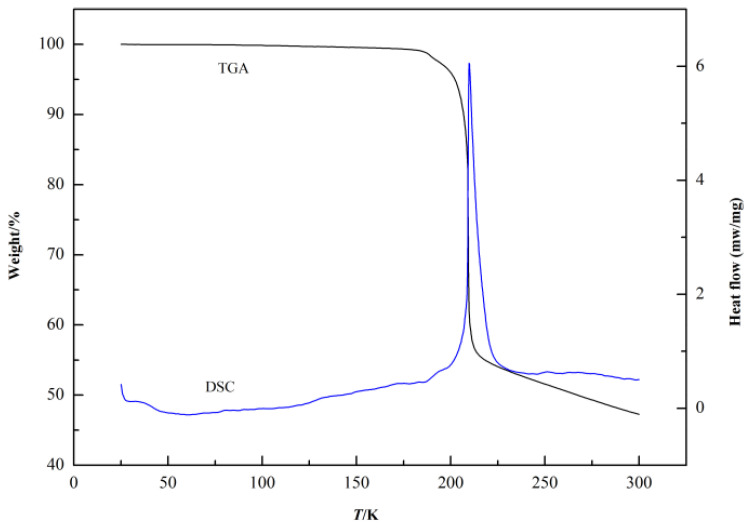
Thermal analysis (TGA/DSC) of thiamine nitrate.

**Figure 4 molecules-28-05012-f004:**
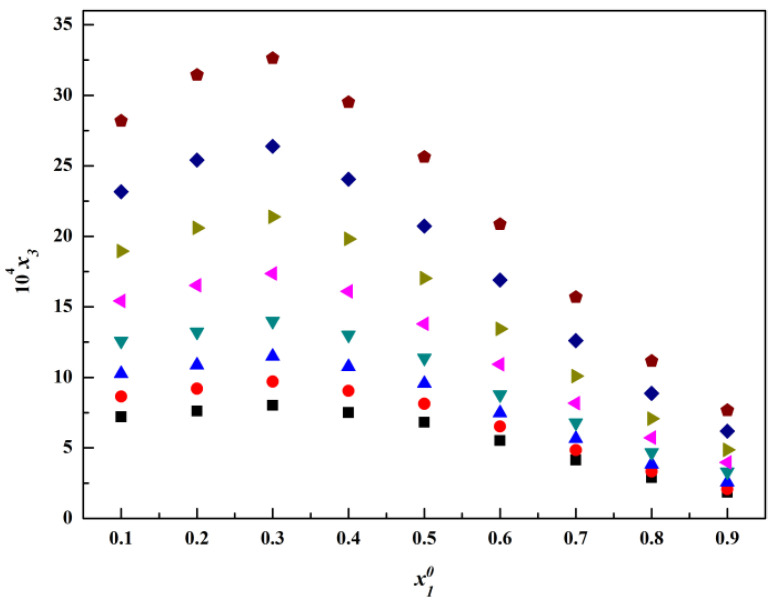
Molar fraction solubility ($x_{3}$) of thiamine nitrate versus molar fraction of methanol ($x_{1}^{0}$) in methanol (1) + water (2) binary solvent 
mixtures at different temperatures: ■, *T
* = 278.15 K; ●, *T* = 283.15 K; ▲, *T* = 288.15 K; ▼, *T* = 293.15 K; ◄, *T* = 298.15 K; ►, *T* = 303.15 K; ◆, *T* = 308.15 K; 
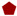
, *T* = 313.15 K.

**Figure 5 molecules-28-05012-f005:**
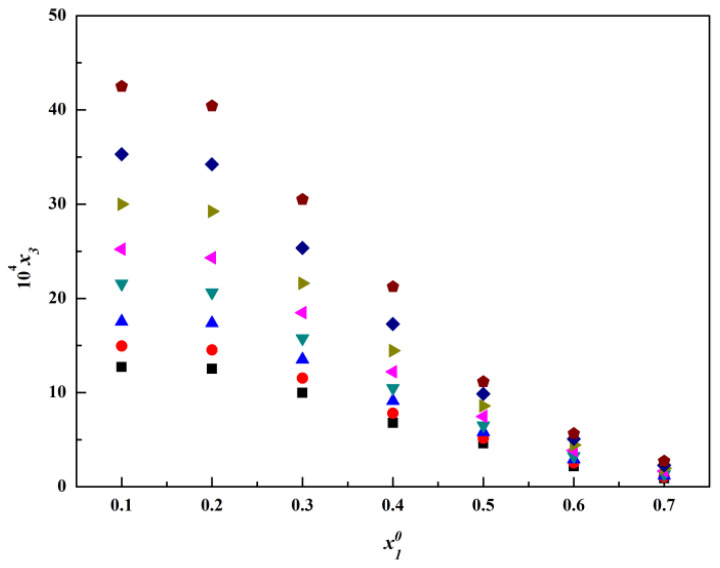
Molar fraction solubility ($x_{3}$) of thiamine nitrate versus molar fraction of acetone ($x_{1}^{0}$) in acetone (1) + water (2) binary solvent mixtures at different temperatures: ■, *T* = 278.15 K; ●, *T* = 283.15 K; ▲, *T* = 288.15 K; ▼, *T* = 293.15 K; ◄, *T* = 298.15 K; ►, *T* = 303.15 K; ◆, *T* = 308.15 K; 
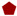
, *T* = 313.15 K.

**Figure 6 molecules-28-05012-f006:**
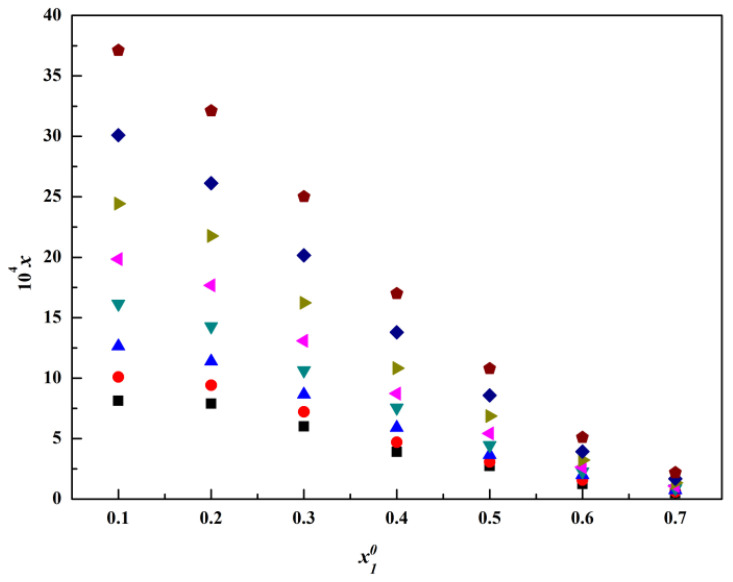
Molar fraction solubility ($x_{3}$) of thiamine nitrate versus molar fraction of isopropanol ($x_{1}^{0}$) in isopropanol (1) + water (2) binary solvent mixtures at different temperatures: ■, *T* = 278.15 K; ●, *T* = 283.15 K; ▲, *T* = 288.15 K; ▼, *T* = 293.15 K; ◄, *T* = 298.15 K; ►, *T* = 303.15 K; ◆, *T* = 308.15 K; 
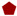
, *T* = 313.15 K.

**Figure 7 molecules-28-05012-f007:**
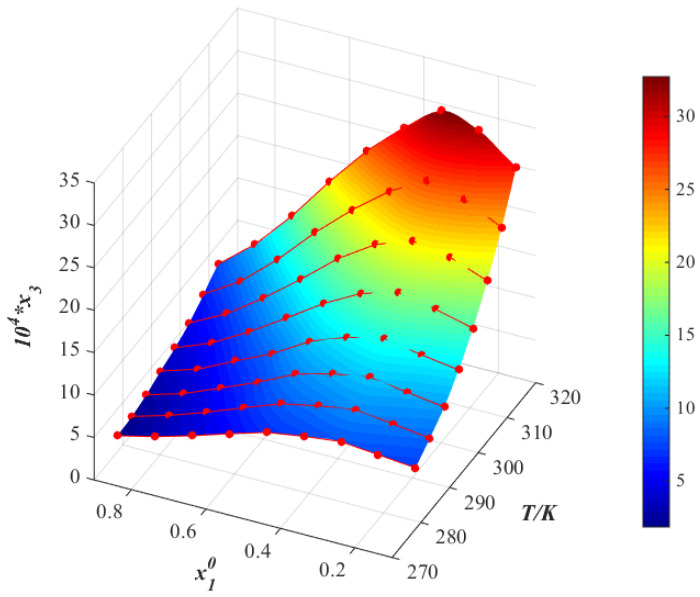
Molar fraction solubility ($x_{3}$) of thiamine nitrate versus molar fraction of methanol ($x_{1}^{0}$) in methanol (1) + water (2) binary solvent mixtures from 278.15 to 313.15 K (*p* = 0.1 MPa). *: The asterisk represents $x_{3}$ times 10^4^.

**Figure 8 molecules-28-05012-f008:**
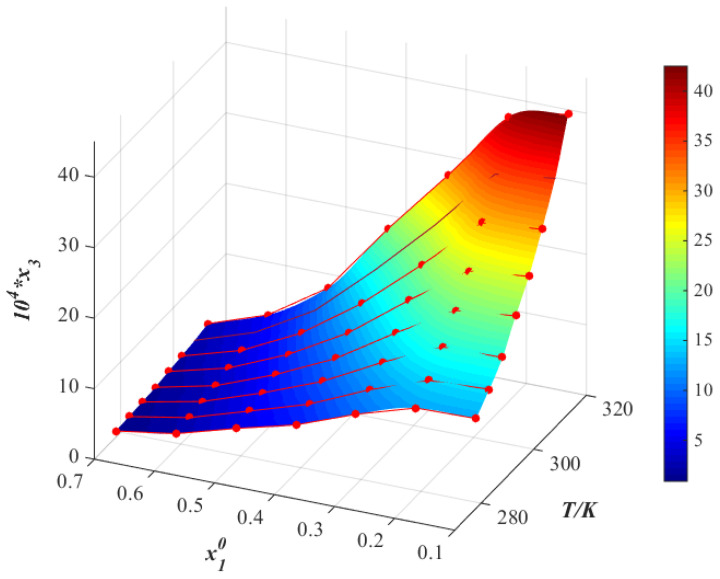
Molar fraction solubility ($x_{3}$) of thiamine nitrate versus molar fraction of acetone ($x_{1}^{0}$) in acetone (1) + water (2) binary solvent mixtures from 278.15 K to 313.15 K (*p* = 0.1 MPa). *: The asterisk represents $x_{3}$ times 10^4^.

**Figure 9 molecules-28-05012-f009:**
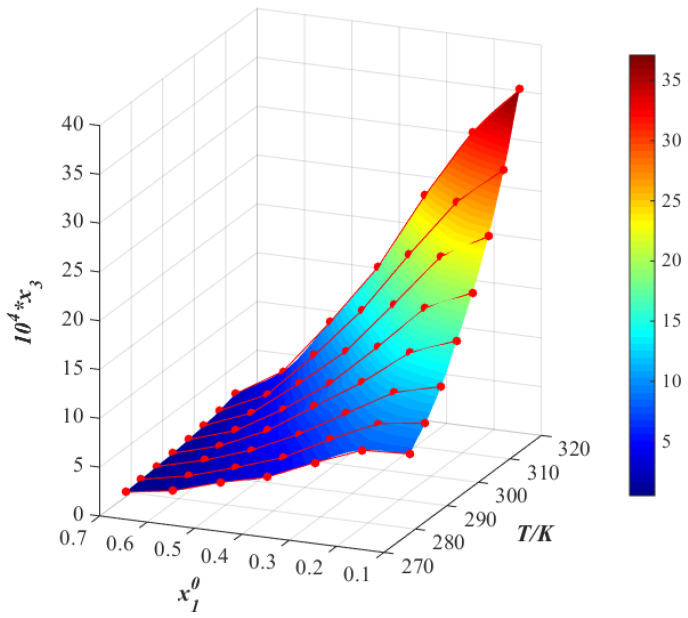
Molar fraction solubility ($x_{3}$) of thiamine nitrate versus molar fraction of isopropanol ($x_{1}^{0}$) in isopropanol (1) + water (2) binary solvent mixtures from 278.15 K to 313.15 K (*p* = 0.1 MPa). *: The asterisk represents $x_{3}$ times 10^4^.

## Data Availability

Not applicable.

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
