# Peer review of "Measurement and Correlation of Solubility of Thiamine Nitrate in Three Binary Solvents and Thermodynamic Properties for the Solutions in Different Binary Mixtures at (278.15–313.15) K"

_molecules, 2023, doi:10.3390/molecules28135012_

Round 1

Reviewer 1 Report

Thiamine nitrate is a member of the vitamin B family, their pharmacological effects has been attracted much attention in recent years. In this work, the solubility of thiamine nitrate in the binary solvent mixtures, including water + methanol, water + acetone and water + isopropanol, was measured, and the experimental solubility was correlated by the modified Apelblat equation, h equation, van’t Hoff equation and NRTL model, respectively. Moreover, the crystal form was verified by using powder X-ray diffraction (PXRD). The study is necessary to establish purification methods and to obtain products with high purity, yields, bulk density and flowability. However, the some revisions should be considered,

1. Experimental measurements under various condition were performed. In general, the dissolving capacity of thiamine nitrate in the binary solvent mixtures at constant temperature ranked as methanol + water > acetone + water > isopropanol + water at the low ratio of water, which is in conformity with the empirical rule “like dissolves like”. While in the high ratio of water systems, the dissolving capacity ranked as water + acetone > water + methanol > water + isopropanol. The occurrence of these phenomena has a complex thermodynamic basis, such as the dielectric constant, ionization constant and surface tension of the different solvents. The mechanism and thermodynamic analyses should be further discussed.

2. The language should be refined and the format of Equations should be corrected according to the style of Molecules.
For example, “p roduce” In Line 64;  
Equations 1, 2, 3, 10, 17, 18, should be placed in appropriate position of the line, similar to other Equations.  
“ (6)”  was lost after Equation 6.

The language should be refined.

Author Response

Response to Reviewer #1

  First of all, we would like to make a grateful acknowledgment to the editor and the reviewers for the valuable suggestions and instructions on improving our manuscript.

We have read the comments and revised the manuscript carefully as follows. All changes are highlighted in the revised manuscript.

Q1. Experimental measurements under various condition were performed. In general, the dissolving capacity of thiamine nitrate in the binary solvent mixtures at constant temperature ranked as methanol + water > acetone + water > isopropanol + water at the low ratio of water, which is in conformity with the empirical rule “like dissolves like”. While in the high ratio of water systems, the dissolving capacity ranked as water + acetone > water + methanol > water + isopropanol. The occurrence of these phenomena has a complex thermodynamic basis, such as the dielectric constant, ionization constant and surface tension of the different solvents. The mechanism and thermodynamic analyses should be further discussed.

A1: Thanks for the suggestion. According to the principle of similar phase dissolution, it can be obtained that the dissolved amount of thiamine nitrate should be: Water + methanol > water + acetone > water + isopropyl alcohol. However, experiments show that as the molar ratio of water decreases, the solubility of thiamine nitrate in water + methanol will increase first and then decrease. Compared with thiamine nitrate in water + acetone and water + isopropyl alcohol, the solubility of thiamine nitrate in water + acetone and water + isopropyl alcohol increases with the molar ratio of water, which is quite different. This also leads to the fact that the solubility of thiamine nitrate in water + methanol is less than that in water + acetone at a high water ratio. We speculate that this is due to the "cosolvency[1]", when water and acetone reach a certain proportion, there will be a peak solubility, which is due to: Due to the hydrogen bond between solute and solvent, organic acetone destroys the three-dimensional hydrogen bond of water, resulting in the occurrence of cosolvency.

[1]Chen, X.; Fadda, H. M.; Aburub, A.; Mishra, D.; Pinal, R. Cosolvency approach for assessing the solubility of drugs in poly (vinylpyrrolidone). Int. J. Pharm. 2015, 494 (1), 346−356.

Q2. The language should be refined and the format of Equations should be corrected according to the styleofMolecules.
Forexample, “p roduce” In Line 64;  

Equations 1, 2, 3, 10, 17, 18, should be placed in appropriate position of the line, similar to other Equations.

 “ (6)”  was lost after Equation 6.

A2: Thanks for the suggestion. I'm sorry for my oversight, the “p roduce” has been corrected. And we modified the format of the equations to meet the standard requirements.

Reviewer 2 Report

Xinda Li et.al reported Measurement and correlation of solubility of thiamine nitrate 2 in three binary solvents and Thermodynamic Properties for the 3 Solutions in different binary mixtures at (278.15-313.15) K

Herein, some scientific and grammatical mistakes are observed. So, I recommend this manuscript for publishing in the Journal molecules with following suggestions:

1.     Author should centralize all the equations

2.     Author should add some quantitative results in abstract

3.     Some sentences are very long and they looked like a paragraph author must split them into two or more sentences as mistake observed in the following sentence “To the best of our knowledge, the solubility of thiamine nitrate in the binary solvent 61 mixtures, including water + methanol, water + acetone and water + isopropanol, has not 62 been reported in the literature, which is the essential data during the solvent screening 63 process to produce thiamine nitrate product with desirable morphology and high yields.”  

4.     Repetition of units is inappropriate mistake observed in the following sentence “the solubility of thiamine nitrate in these binary solvent mixtures was meas- 65 ured over the temperature range from 278.15 K to 313.15 K” corrected form is 278.15 to 313.15 K.

5.     Concise the conclusion as it is too much lengthy

6.     Author should explain what is “DSC”.

7.     Overall language of manuscript needs improvement

8.     This sentence need correction “Because of the polarity of the molecular, the morphology of thiamine nitrate is often like needle or rod”, author must write it as “Due to molecular polarity  the morphology of thiamine nitrate is often needle or rod like”.

9.     The correct unit of resistivity is “Ωm. Author must correct it mistake observed in the following sentence “Distilled-deionized water with resistivity of 18.2 MΩ cm

10.  What author actually want to explain in this sentence “SP3 is the number of non-ring, nonterminal SP3 atoms, SP2 is the number of non- 237 ring, nonterminal SP2 atoms and RING represents the number of independent single, 238 fused, or conjugated ring systems” author must correct it.

Extensive editing of English language required

Author Response

Response to Reviewer# 2

First of all, we would like to make a grateful acknowledgment to the editor and the reviewers for the valuable suggestions and instructions on improving our manuscript.

We have read the comments and revised the manuscript carefully as follows. All changes are highlighted in the revised manuscript.

Q1. Author should centralize all the equations

A3: Thanks for the constructive comment. We have put the equations together and put it in the attachment.

Q2.  Author should add some quantitative results in abstract

A4: Thanks for the constructive comment. We have made modifications in the abstract, adding the mole fraction of water with high water ratio, giving the trend of solubility change under different measurement conditions in the abstract, and the actual measurement data are given in the main paper.

Q3. Some sentences are very long and they looked like a paragraph author must split them into two or more sentences as mistake observed in the following sentence “To the best of our knowledge, the solubility of thiamine nitrate in the binary solvent 61 mixtures, including water + methanol, water + acetone and water + isopropanol, has not 62 been reported in the literature, which is the essential data during the solvent screening 63 process to produce thiamine nitrate product with desirable morphology and high yields.”

A5: Thanks for the suggestion. We have changed the paragraph to make it more smooth, please see paragraphs 61-64 for specific changes.

Q4.  Repetition of units is inappropriate mistake observed in the following sentence “the solubility of thiamine nitrate in these binary solvent mixtures was meas- 65 ured over the temperature range from 278.15 K to 313.15 K” corrected form is 278.15 to 313.15 K.

A4: Thanks for the suggestion. We have changed the “from 278.15 K to 313.15 K” in the article to “from 278.15 to 313.15 K”, and the modified content is shown in the header and line 65.

Q5.  Concise the conclusion as it is too much lengthy

A5: Thanks for the suggestion. We have rewritten the conclusion to make it shorter, please refer to the conclusion for specific modifications.

Q6.  Author should explain what is “DSC”.

A6: Thanks for the suggestion. The TGA/DSC is explained in section 2.4

Q7.  Overall language of manuscript needs improvement

A7: Thanks for the suggestion. The Writing errors you mentioned above have been revised in the resubmitted manuscript. Furthermore, the English writing of the whole manuscript has been carefully rechecked and corrected in the revised manuscript.

Q8.  This sentence need correction “Because of the polarity of the molecular, the morphology of thiamine nitrate is often like needle or rod”, author must write it as “Due to molecular polarity  the morphology of thiamine nitrate is often needle or rod like”

A8: Thanks for the suggestion. We have changed the “Because of the polarity of the molecular, the morphology of thiamine nitrate is often like needle or rod” in the article to “Due to molecular polarity the morphology of thiamine nitrate is often needle or rod like”, and the modified content is shown in line 48.

Q9.  The correct unit of resistivity is “Ωm”. Author must correct it mistake observed in the following sentence “Distilled-deionized water with resistivity of 18.2 MΩ cm”

A9: Thanks for the suggestion. We have changed the “Distilled-deionized water with resistivity of 18.2 MΩ cm” in the article to “Distilled-deionized water with resistivity of 18.2 Ωm ”, and the modified content is shown in line 78.

Q10.  What author actually want to explain in this sentence “SP3 is the number of non-ring, nonterminal SP3 atoms, SP2 is the number of non- 237 ring, nonterminal SP2 atoms and RING represents the number of independent single, 238 fused, or conjugated ring systems” author must correct it.

A10: Thanks for the suggestion. We have reinterpreted the comments on the equation to make it more smooth, and the modified content is shown in line 244-245.

Reviewer 3 Report

The solubility of compounds in different solvents is a basic and important issue in physical chemistry. In the present paper, the authors investigate the solubility of thiamine nitrate in various water-containing binary mixed solvents and some unreported insights. In my opinion, the results presented here are generally interesting and are deserved to be published on Molecules. I have some suggestions as follows:  

1.In Figure 2, the intensity values are meaningless because those different XRD patterns are compared in one figure. So, I recommend removing them. In addition, is it possible to add a simulated XRD pattern sourcing from the single-crystal XRD result, which is helpful to ensure the structure of the prepared compound?

2. For the TGA curve of thiamine nitrate, an analysis of weight loss is necessary, which is not mentioned. In addition, the highest heating temperature is too low, and 800 ℃ is generally expected.

Author Response

Response to Reviewer # 3

First of all, we would like to make a grateful acknowledgment to the editor and the reviewers for the valuable suggestions and instructions on improving our manuscript.

We have read the comments and revised the manuscript carefully as follows. All changes are highlighted in the revised manuscript.

Q1: In Figure 2, the intensity values are meaningless because those different XRD patterns are compared in one figure. So, I recommend removing them. In addition, is it possible to add a simulated XRD pattern sourcing from the single-crystal XRD result, which is helpful to ensure the structure of the prepared compound?

A1: Thanks for the suggestion. We have removed the intensity values of the peaks in the XRD pattern.

The simulated XRD pattern sourcing from the single-crystal XRD result is shown in the figure 2(b). All the characteristic peaks of single-crystal XRD pattern can be found in the X-ray powder diffraction (PXRD) pattern (figure 2(a)), and thus the structure of the prepared compound was similar to thiamine nitrate single-crystal.

Figure 2. (a) Powder X-ray diffraction (PXRD) patterns for excess solid of thiamine nitrate in different conditions: (a = raw materials; b, c, d, e = sediments in binary solvents methanol (  = 0.10, 0.30, 0.50, 0.70, 0.90) + water at T = 298.15 K, respectively). (b) The simulated XRD pattern sourcing from the single-crystal XRD.

Q2: For the TGA curve of thiamine nitrate, an analysis of weight loss is necessary, which is not mentioned. In addition, the highest heating temperature is too low, and 800 â„ƒ is generally expected.

A2: Thanks for the suggestion. At present, the TGA of thiamine nitrate has been reported in related articles, which will lose weight rapidly at 195℃ and have a large exothermic peak, and lose weight slowly at 208℃. In addition, according to literature reports, according to DSC measurement, there is a strong exothermic peak at 198℃, and TGA shows that the mass decreases at this time, indicating that thiamine nitrate begins to decompose before melting, and DSC shows that this temperature is the decomposition peak, which means that DSC cannot be used to calculate its melting enthalpy, so the heating temperature in this paper does not reach 800℃. Just to show that you can't use DSC to get the enthalpy of melting.
